# South African Healthcare Professionals’ Knowledge, Attitudes, and Practices Regarding Environmental Sustainability in Healthcare: A Mixed-Methods Study

**DOI:** 10.3390/ijerph191610121

**Published:** 2022-08-16

**Authors:** Helga E. Lister, Karien Mostert, Tanita Botha, Simoné van der Linde, Elaine van Wyk, Su-Ané Rocher, Richelle Laing, Lucy Wu, Selma Müller, Alexander des Tombe, Tebogo Kganyago, Nonhlanhla Zwane, Boitumelo Mphogo, Filip Maric

**Affiliations:** 1Department of Occupational Therapy, Faculty of Health Sciences, University of Pretoria, Pretoria 0028, South Africa; 2Department of Physiotherapy, Faculty of Health Sciences, University of Pretoria, Pretoria 0028, South Africa; 3Department of Statistics, Faculty of Natural and Agricultural Sciences, University of Pretoria, Pretoria 0028, South Africa; 4Department of Health and Care Sciences, Faculty of Health Sciences, UiT The Arctic University of Norway, 9020 Tromsø, Norway

**Keywords:** environmental sustainability, sustainable healthcare, occupational therapy, physiotherapy, planetary health, climate change, healthcare professions

## Abstract

Climate change, biodiversity loss and large-scale environmental degradation are widely recognized as the biggest health threats of the 21st century, with the African continent already amongst the most severely affected and vulnerable to their further progression. The healthcare system’s contribution to climate change and environmental degradation requires healthcare professionals to address environmental issues urgently. However, the foundation for context-relevant interventions across research, practice, and education is not readily available. Therefore, we conducted a convergent mixed-methods study to investigate South African healthcare professionals’ knowledge, attitudes, practices, and barriers to environmental sustainability. Healthcare professionals participated in a cross-sectional questionnaire (*n* = 100) and in-depth semi-structured focus group discussions (*n* = 18). Data were analyzed using descriptive statistics and thematic analysis, respectively, and integrated to provide holistic findings. Our results confirm overwhelmingly positive attitudes and a high degree of interest in education, implementation, and taking on more corresponding responsibility, but a lack of substantial knowledge of the subject matter, and only tentative implementation of practices. Identified barriers include a lack of knowledge, resources, and policies. Further research, education, and policy development on overcoming these barriers is required. This will facilitate harnessing the extant enthusiasm and advance environmental sustainability in South Africa’s healthcare practice.

## 1. Introduction

Climate change, biodiversity loss and large-scale environmental degradation are widely recognized as the biggest threats to 21st century health and civilizations [1,2,3]. The African continent is amongst the most affected by climate change and environmental degradation and is most vulnerable to their further progression. Climate change projections for Sub-Saharan and Southern Africa show a clear warming trend in inland subtropics, alongside a higher frequency of extreme heat events, increasing aridity, precipitation changes, sea-level rise, and extensive biodiversity loss [4,5,6,7,8].

With Sub-Saharan Africa remaining the poorest region in the world, these environmental changes have grave implications for its people and a large proportion of low-income communities [9,10]. Though knowledge about the precise pathways by which environmental change affects health needs further refinement [11], it is evident that it has a variety of significant, adverse social and health impacts. Amongst other issues, climate change, biodiversity loss, deforestation, desertification, droughts, reducing water quality and resources, and land degradation are increasing the prevalence of malaria and other food-, water-, and vector-borne diseases in areas where they have previously not been present; driving soil impoverishment and reduced agricultural output that will further challenge food insecurity and undernutrition; increasing migration, urban poverty and environmental injustice that comes with rapid urbanization in the region; threatening child development and increasing susceptibility to diseases; driving extreme weather event-related fatalities and injuries; and increasing the likelihood of heat stress, heat stroke and death due to prolonged exposure to high temperatures [5,6,7,8,12,13,14,15,16,17].

In addition to these health impacts, the urgency for healthcare professionals to address environmental issues, directly and indirectly, is further exacerbated by the fact that healthcare systems and services contribute to climate change and environmental degradation. Healthcare-related carbon emissions, waste production, plastic pollution, toxic waste, and other undesirable outputs are also threatening good health, or even contributing to poor health in a Sisyphean process of improving the health of people [18,19,20]. Nevertheless, neither contribution to the problem nor contribution relative to the size of the country, healthcare system, or service are necessary to recognize the ethical imperative and professional stringency for the medical and health professions to understand and act on biodiversity loss, climate change and environmental degradation. The fact that these affect people’s health and the possibility of receiving or delivering better healthcare alone is sufficient to recognize the urgent need for action on planetary health and sustainable healthcare worldwide [2,13,21].

The World Health Organization (WHO) recently highlighted four urgent actions for the health community: (1) the training of the health workforce to respond to climate change, (2) taking climate action in the healthcare sector, (3) enabling health professional advocacy on climate change and health, and (4) protecting the health of future generations [3]. Increasing numbers of health professionals around the world are becoming cognizant of the link between human and environmental health [22,23,24,25,26]. This is evident in the following: climate change, sustainability and related issues are integrated gradually into relevant policy frameworks, guidelines, and education programs [27,28,29]; new approaches to healthcare practice are being discussed and developed within and outside of traditional clinical contexts; and increasing efforts are going into the reduction in healthcare’s environmental footprint [30,31,32,33].

In the context of Sub-Saharan and Southern Africa, improvements in the sustainability of healthcare systems have been primarily focused on social and economic aspects, as necessitated by considerable socioeconomic and political challenges [4,34,35]. However, action on planetary health and environmental sustainability is foundational to truly sustainable healthcare [8,36]. To date, few reported efforts are improving Southern African healthcare systems and services’ environmental sustainability. There remains limited insight into healthcare professionals’ knowledge, attitudes, and practices (KAP) regarding environmental sustainability [4,37,38,39]. A better understanding of South Africa’s healthcare professionals’ KAP regarding environmental sustainability is urgently needed to make context-relevant suggestions for corresponding intervention in healthcare research, practice, and education.

In this article, we present findings from a convergent mixed-methods study, combining a cross-sectional questionnaire and semi-structured focus group discussions (FGDs) to identify South African healthcare professionals’ KAP and barriers to environmental sustainability. Our findings point to a pressing need and desire for: (1) healthcare professional education in environmental sustainability, (2) factoring environmental sustainability into clinical practice guidelines while ensuring availability of matching equipment and infrastructure, and (3) corresponding policy development. Timely research, development, and implementation across these three arenas will be essential to advance the implementation of environmental sustainability into South African healthcare systems and daily professional practice.

## 2. Methods

### 2.1. Study Design

To determine the KAP of the selected healthcare professionals, a convergent mixed-methods design was executed, consisting of two parallel phases of data collection. One phase was quantitative, and the other was qualitative [40]. During the quantitative phase, a questionnaire, we collected information on the self-reported knowledge about, attitudes towards and practices in terms of environmental sustainability in healthcare. In the qualitative phase, we gathered data during FGDs using open-ended questions. From the deductive analysis of the transcripts of these discussions, themes emerged which reflected the KAP of the participants.

We completed separate data analyses and brought the findings together to present a comprehensive view. We discuss each phase’s methodology and the results’ convergence below.

### 2.2. Quantitative Phase

#### 2.2.1. Quantitative Research Design

We conducted a cross-sectional questionnaire in this phase. The following healthcare professionals were invited to participate: audiologists, human nutritionists, occupational therapists, physiotherapists and speech-language pathologists. The professional had to be a South African citizen registered with the Health Professions Council of South Africa (HPCSA). The target population included clinicians and academics with at least six months of working experience since completing a year of compulsory community service in the public sector. The final analysis included a convenience sample of 100 healthcare professionals spanning several professions, regions and universities attended. The authors identified participants in their networks, who shared it again with their communities of practice.

#### 2.2.2. Data Collection

Data were collected after we received ethical clearance from the Ethics Committee, Faculty of Health Sciences, the University of Pretoria (reference number 798/2020). The questionnaire was developed based on a literature review about environmental sustainability in healthcare and consisted of five sections: (1) demographics; (2) participants’ knowledge about local and global environmental issues and their health impacts; (3) attitudes towards environmental sustainability in general and concerning healthcare practice within their respective fields; (4) current practices in their respective professional contexts to maintain a healthy environment; and (5) barriers in implementing environmental sustainability and healthcare practice (See Appendix A).

In Section 2, participants had to match six definitions with three terms, and score their level of self-perceived knowledge on a 4-point Likert scale. In Section 3, they had to indicate their level of agreement with different statements on a 5-point Likert scale (‘ranging from strongly agree’, to ‘strongly disagree’). In Section 4, participants indicated (‘yes or no’) whether they implemented environmentally sustainable practices. In Section 5, participants could select from a list what may be hindering their implementation of environmental sustainability into their practice and questions regarding educational input. Open-ended questions complemented Section 2, Section 4 and Section 5.

A senior research consultant and statistical analyst at the University of Pretoria, as well as two occupational therapists and three physiotherapists with a particular interest in environmental sustainability, gave feedback on the questionnaire to ensure face validity. We uploaded the questionnaire to the electronic platform Qualtrics. Finally, we distributed the questionnaire as an anonymous link via emails and using social media.

#### 2.2.3. Data Analysis

Two hundred and three participants started the questionnaire; however, only 100 completed it. The participants’ results with 100% questionnaire completion were considered and analyzed. We used descriptive statistics to describe the demographics of the participants. The summary statistics are given in the form of frequency tables for categorical variables and descriptive statistics such as mean, range, standard deviation for numeric answers.

### 2.3. Qualitative Phase

#### 2.3.1. Qualitative Design

We used an exploratory, descriptive study design to qualitatively investigate healthcare professionals’ perceptions and practices regarding environmental sustainability [41]. The population for this study phase were registered South African healthcare professionals, namely audiologists, occupational therapists, physiotherapists, and speech-language pathologists. Participants had to be registered with the Health Professions Council of South Africa (HPCSA) and came from private, non-governmental, public, and academic settings. We used purposive sampling to identify possible individuals willing to participate in FGDs. Participants were identified via the researchers’ networks. Snowball sampling followed when identified participants, in turn, suggested potential future participants. At the end of each FGD, the participants were asked to send us details of other possible participants.

#### 2.3.2. Data Collection

Data were collected after we received ethical clearance from the Ethics Committee, Faculty of Health Sciences, the University of Pretoria (reference number 559/2020). Participants were contacted via email. The FGDs were conducted on the informed consent brochures, and the demographic section of the questionnaire was emailed to participants to complete and return. The demographic questionnaire collected information about participant’s date of birth, sex, profession, work sector, work setting, year of qualification, type of qualification, and university at which they graduated. The guest link to the online session on the Blackboard Collaborate Ultra (BCU) platform was distributed to participants via email. Five FGDs were conducted with the number of participants per session ranging from three to five [42]. A semi-structured FGD schedule guided the discussion. The opening question was, “What do you understand by ‘environmental sustainability’?” The sessions lasted an average of 40 min. All FGD sessions were audio recorded. Seven of the authors (SvdL, SR, LW, AdT, TK, NZ and BM) transcribed the recordings verbatim, where after and each checked a different transcript for accuracy.

#### 2.3.3. Data Analysis

The transcripts were analyzed in six steps common to thematic analysis [43,44]: (1) The first step was to familiarize ourselves with the data, by reading and re-reading the transcripts. (2) Hereafter, units of meaning were labelled with codes. Next, similar codes were categorized together. (3) Similarly, these categories were organized into themes. (4) The themes were then reviewed against the codes and the data set. (5) The next step was naming the themes before, finally, (6) selecting quotes from the original codes that best supported the emergent themes.

### 2.4. Integrated Analysis

After the data from the quantitative and qualitative phases were analyzed separately, the results were compared, and congruences and differences were determined. These are presented as descriptive statistics in tables and narrative passages in the following section.

## 3. Results

### 3.1. Demographic Characteristics

The participants of the quantitative phase were mainly from the OT profession (69%), whereas in the qualitative phase, the majority belonged to the PT profession (11/18 participants). Age ranges for participants in both phases were very similar, with a mean of 33.6 and 39.9 for the quantitative and qualitative phase, respectively. Please see Table 1 below for further details. 

In the quantitative phase, most (31%) of the participants’ primary area of work was within the private practice sector. The area with the second most participants was the education sector, comprising 15% of the population. The other main work areas include mental health services, community, and acute or secondary care. Table 2 for the quantitative phase shows more than 100 responses to the primary area of work because the participants were allowed to choose more than one option of where they were currently working.

In the qualitative phase, the number of participants was evenly distributed between private and public sectors, with a few from non-governmental organizations. Participants were mainly clinicians and educators compared to academics/researchers and managers. A number of the participants in the qualitative phase were also junior physiotherapists in the public sector who rotated between settings. Work settings were evenly distributed between private practices, public hospitals, and local authorities.

### 3.2. Knowledge

In the quantitative phase, most participants (83–87%) correctly defined essential terms such as sustainable healthcare, environmental sustainability, and environmental degradation. When asked whether they consider environmental sustainability in practice, 51% of the participants reported that they did not. They also indicated that they would describe their current knowledge regarding environmental sustainability as limited. They described their knowledge specific to environmental degradation, sustainability, and sustainable healthcare as limited. Of the participants, 77% reported that they believed that environmental degradation affects disease patterns and human health. The remainder indicated that they did not know if they believed that environmental degradation directly influences healthcare (Table 3).

In the qualitative phase, the central theme regarding the participant’s knowledge of environmental sustainability revolved around the insight that human well-being is dependent on a healthy environment. Participants emphasized that *“the basic needs for human life are reliant on the environment being able to sustain and produce itself”* (participant 13, PT, female); and that *“environmental sustainability is highly related to human wellness because we rely on the environment for daily living”* (participant 13, PT, female).

Beyond mere necessity for survival, participants also noted that the natural world can have positive effects on human well-being that need to be safeguarded for future generations, as *“let**’s say a bird singing and good oxygen and the wind in the willows, that is good for the human, that**’s something we would like to sustain that is good for future generations”* (participant 7, PT, female).

Negatively speaking, both deleterious physical and mental health effects were discussed by our participants, alongside the need to avoid these: *“If we just look into the respiratory kind of a shift that* [climate change] *can have on a patient… If there* [is] *dust, for instance,… and dust levels that**’s been controlled without having an effect on your patients’* [clients’] *health, …the physical, as well as the psychological health of the patient”* (participant 5, PT, female); *“…but let**’s say the* [degraded] *environment* [can influence] *cancers and sinusitis and problems with new-born babies. We don**’t want… that”* (participant 7, PT, female).

Living in informal settlements that remain highly common in Southern Africa, that is, urban areas where inhabitants have small self-made dwellings often made of iron sheets, cardboard, or wood, was also highlighted as being detrimental to healthcare due to poor environmental conditions: *“One big area that really influences health everywhere, and all our patients* [clients] [is that] *in* [informal settlements they] *are living in conditions, that**’s not conducive to good health. So, it always impacts on their health. And again, psychological as well. If you don**’t have a clean environment, a nice place to live, and clean air to breathe, of course, you will get sick”* (participant 4, PT female).

Building on the insight that human health and well-being depend on conducive environmental conditions, which, therefore, warrant protection, environmental sustainability was principally discussed as a question of managing resource consumption to, quite in-line with common definitions, ensure their availability to present and future generations alike. As noted by participant 6, PT, female, it is *“anything that we can do to use our resources sparingly, making sure that we leave something full for the future”*.

### 3.3. Attitudes

In the quantitative phase, 91% participants agreed that the current state of environmental degradation is concerning, with 5% disagreeing and 4% being neutral on the matter. Similarly, 91% percent of the participants also agreed that environmental sustainability should be incorporated into healthcare, as opposed to 4% who disagreed and 5% who neither agreed nor disagreed. That “the lack of environmental sustainability in the healthcare sector has an impact on our environment” was also agreed on by 88% of the participants, while 5% disagreed and 7% were impartial. Additionally, equally, 91% of the participants supported the statement that “environmental degradation has an impact on the health of individuals throughout their lifetime” (50% strongly agree and 41% agree) while only 4% of the participants disagreed.

Most of the participants felt that healthcare professionals should take a leading role in advocating and implementing environmental sustainability in the healthcare sector (57% agree and 27% strongly agree), while only 7% disagreed with this. This finding differed slightly to questions of responsibility for environmental sustainability in the healthcare sector. Here, the highest percentage of participants felt that the government (Department of Health) should take responsibility (92%), followed by healthcare professionals (89%), leadership in communities (counsellors, chiefs, etc.) (84%) and individuals (clients) (80%). Yet, the absence of significant difference between these percentages, indicated a relatively even share of responsibility in the eyes of our participants.

Although the participants reported a relative lack of knowledge and information regarding environmental sustainability in healthcare practice, an overwhelming 97% indicated that they would be interested in implementing relevant strategies and 93% were interested in corresponding educational input, with only 2% opposed and 5% impartial. 

The high degree of interest in implementation and education was also matched in the FGDs, but was additionally linked to passing on awareness, knowledge and practice to patients [clients], as to *“…reduce the impact on the environment* [which] *is at this stage about education and informing people”* (participant 5, PT, female); *“you have to educate them on what is environmental sustainability, how can the environment be preserved and what are the benefits of as well as making them understand”* (participant 13, PT, female); *“…obviously as a professional creating more awareness, like using less paper in our practices and like the more practical things so I know like with nowadays like of the more telehealth is you know a thing coming up”* (participant 3, OT, female); *“we have to start working on creating more awareness and making people aware how to use things, and reuse things”* (participant 2, OT, female).

Matching general knowledge of human health depending on environmental integrity, concerns for future generations and the benefits of looking after the environment were also expressed clearly by participant 4, PT, female: *“Basically, being more friendly to the environment so that future generations can also have time on earth experiencing the environment as it is today, and still experiencing nature, as it is.”*

### 3.4. Practices

In the quantitative phase, more than half of the participants (58%) disagreed with the statement that “the healthcare sector is currently taking environmental sustainability into consideration in healthcare practices”, while only 8% of the participants agreed and 19% remained impartial. The same number of participants (58%) also reported that they had considered implementing environmental sustainability in practice. More specifically, 44% reported that they were implementing environmental sustainability in some form, while 51% claimed not considering it yet when using or purchasing equipment, resources, consumables, or other devices.

In the qualitative phase, three main practice themes emerged from the analysis, covering: intervention and practice environments; practices related to waste, recycling, reusing, and reducing the use of resources; and opportunities for advancing environmental sustainability.

#### 3.4.1. Theme 1: Intervention and Practice Environments

In the FGDs, it became clear that some participants were in favor of clients spending time outdoors during treatment sessions and when relaxing: *“Let**’s say it**’s after a stroke and rehabilitation will be lifelong in an ageing population. My feeling is* [that] *if you can combine nature with the rehabilitation, that is better”* (participant 7, PT, female); *“…encourage people to grow plants, or if you cycle, that is, encouraging patients* [clients] *to grow their own tree and doing a vegetable garden… and reducing our carbon dioxide footprint”* (participant 9, speech therapist, female); *“at an old age home we gave palliative care… but we were still be able to take them outside and they used to really enjoy going to the sun”* (participant 1, PT, male); *“working in public health having group classes outside benefits the patient* [clients] *a lot more; they seem to be a lot more cheerful. Even if we look at psychological and mental aspects of rehab* [rehabilitation] *… outdoors they definitely have more of a positive attitude”* (participant 1, PT, male).

The benefits of outdoors practice were further supported by the observation that practice environments can also be less than favorable: *“The department where I used to work was not very neat and there were old walking frames standing around that needed to be repaired”* (participant 14, PT, female).

#### 3.4.2. Theme 2: Practices Related to Waste, Recycling, Reusing and Reducing the Use of Resources

There was a matching focus on waste and practices related to recycling, reusing, or reducing the use of wasteful materials across the quantitative and qualitative phase of our study. In the quantitative phase, 24% of the participant that chose to elaborate on the open-ended questions stated that they were currently recycling in their practice, and 7% mentioned using biodegradable materials in practice. Additionally, 73% indicated that they consider reusing and recycling when buying/using equipment and resources, 9% indicated considering biodegradable materials, and another 9% consider the use of less equipment. Participants also indicated that they thought that equipment such as PPE, thermoplastic and plastic packaging, paper, and plastic equipment contribute to climate change and environmental degradation. 

In the qualitative phase, participants reported on recycling in hospitals, departments, practices, and in their capacity, but also noted inconsistencies in implementation: *“we use our recyclable materials to do activities, especially in hospitals on areas or community areas where there is not a lot of resources”* (participant 5, OT, female); *“…there are active recycling projects happening at the hospitals. However, I find that most of the hospital’s important waste, like paper that could be recycled, is just thrown into normal dustbins”* (participant 13, PT, female). *“…with COVID-19 at this stage we are just throwing away so much plastic and gloves and aprons and stuff it actually breaks your heart”* (participant 7, PT, female).

To some extent, these inconsistencies or tentativeness were also mirrored in more personal attempts and reflections on implementing environmentally sustainable practice. This was clearly expressed by participant 6, PT, female who stated: *“I try to implement it into both my professional and personal life, but unfortunately you can’t really be equally environmentally sustainable or environmentally conscious in your personal and professional life if you are a healthcare worker”* (participant 6, PT, female).

Giving a few more specific examples, participant 7, PT, shared her efforts in moving to paperless work, using fewer electronic apparatuses, solar heating, and saving electricity: *“I must say going from paperless to the cloud... I prefer my accounts online. But my clinical data is on paper”*; *“we have got solar heating on the premises where the practice is”*; *“I think the load shedding made a lot of physios think about using machines. If a patient expects a certain method of treatment and it’s not possible to give it; that they might cancel appointments or feel that they are not properly attended to. The way that I work, I never touch a machine, although I have an electrically adjustable plinth that I absolutely love”* (participant 7, PT, female).

#### 3.4.3. Theme 3: Opportunities for Advancing Environmental Sustainability

The discussions also made evident that participants saw considerable potential for advancing environmental sustainability within the South African context: *“So I think there**’s like so many opportunities and potential in South Africa that we can use to make use of, like, recycling to benefit the community as well”* (Participant 3, OT, female); *“and the thing about Africans, we have so many opportunities like putting solar panels for hospitals to save on the energy they need”* (participant 2, OT, female).

A significant opportunity noted was also the potential to act as role models in environmental sustainability: *“Physios especially in private practice who have a lot of people they interact with, and people always see healthcare professionals as sort of role models, you, if they see people are doing it then it**’s not that much of an effort, and it is the right thing to do, then they will be more eager to try it themselves. So, I think we have to be an example, you know just to make the difference, where you are in your own practice, and then be like a role model or a champion for environmental sustainability”* (participant 4, PT, female).

### 3.5. Barriers

In the quantitative questionnaire, participants were able to choose more than one barrier from a pre-determined list; as well as indicate other barriers not listed with a description. From this, the main barriers for the implementation of environmental sustainability included a lack of knowledge (*n* = 64, 37%), a lack of finances (*n* = 38, 22%), the effort required (*n* = 20, 12%), a lack of time (*n* = 18, 10%) and a lack of resources (*n* = 18, 10%). Only two persons (1%) indicated that they were not interested in having an environmentally sustainable practice. Additional information provided included the poor accessibility of environmentally sustainable materials and equipment, and the costs associated with purchasing these; as well as the influence of team dynamics such as a lack of collective insight, common goals and buy-in from all the members.

In the qualitative phase, the barriers discussed all related to either the lack of relevant policies or infrastructure. As argued by participant 2, OT, female, *“…in the current environment we don**’t really have many of those recycle… policies in place”* (participant 2, OT, female). Closely related to that the absence of recycling depots and collections by the local authority, even in urban areas, hindered good intentions, especially as it could be at the person’s own cost: *“You really have to drive from facility to facility with bags in the back of your boot during office hours to recycle. So as much as we would like to do it is a difficult thing”* (participant 7, PT, female); *“So, if it**’s made easier, then people will probably do it more and will be more positive about it”* (participant 4, PT, female).

## 4. Discussion

The findings from our study of South African health professionals’ KAP regarding environmental sustainability in healthcare strongly resonate with findings from similar studies from other regions of the world [22,23,26]. More specifically, our study confirms that there is general awareness of the subject matter and some relevant terms, alongside basic recognition of the dependence of human life on environmental integrity. Yet, there is a lack of substantial theoretical and practical knowledge, which was also perceived as such by study participants, and likely contributed to the only tentative implementation of relevant practices related to waste reduction, recycling, reusing, or reducing the overall amount of resource consumption. These efforts, however, were far from clearly or formally implemented in either organizational contexts or personal practice and appear impeded by a host of considerable barriers. In addition to lacking knowledge, barriers include lack of financial and material resources, time, and relevant policies and infrastructures that would embed and facilitate implementation in South African healthcare practice.

What is encouraging though, is that most participant health professionals considered environmental sustainability and its further implementation as being of critical importance for the health of present and future generations alike. While many expressed that responsibility for its advancement in the health sector should be shared among governments, health professions, communities, and clients the vast majority of study participants also shared the sense that healthcare professionals should show much more involvement and leadership in this area and act as role models to clients and communities, based on a growing understanding of the relationships between health and environment. This largely positive trend can be observed clearly everywhere now, as relatively new fields such as sustainable healthcare and planetary health are growing at a rate that is making it near impossible to track the waves of new publications, research projects, organizations, activist efforts, education initiatives, etc., that are emerging everywhere around the world.

That our study asserts lack of knowledge as one of the main barriers to advancing environmental sustainability in healthcare [45] simultaneously singles out education as one of the most central strategies for decreasing this lack and so laying the foundation for subsequent implementation. To this end, our study justifies the considerable efforts that are now going into planetary health and sustainable healthcare education around the world and include the development of relevant education frameworks, competencies, content, and education formats [27,28,46,47,48]. In the South African context, however, very little integration of planetary health and sustainable healthcare into medical and healthcare professional curricula is happening to date, thus demarcating a clear and urgent field for action and corresponding research that should encompass everything from fundamental understandings about the relationships between health and environment all the way through concrete strategies for implementation in daily practice. This is confirmed in another recent interview study with key South African experts [49].

While lack of finances was identified as the other main barrier next to lack of knowledge, the fact that this was associated particularly related to equipment accessibility and costs potentially also points to an issue that might nonetheless be remedied by way of education (and related research and innovation). That is, if the high cost of more environmentally sustainable equipment is perceived as a barrier, it begs to question whether more education about the reduce-reuse-recycle triad (and maybe also its ‘rethink’ extension) might not be critical to overcome the perception that equipment itself might be the solution, rather than its reduction, reuse, and recycling [45]. This is of particular relevance because South Africa’s public healthcare sector remains under-resourced [50].

As highlighted by our study participants, and equally in-line with other studies, however, it should be clear that education is far from a silver bullet since a range of systemic barriers are significantly impeding the implementation of environmental sustainability in South African healthcare practice across education, individual practitioner, and systems scale [51,52,53]. Beyond a lack of finances then, these barriers include a lack of time, material resources, and, critically, a lack of policies and relevant infrastructure that would embed and facilitate environmental sustainability in South Africa’s healthcare practice, also by addressing the need for education, financing, time, and material resources, and thus lowering the threshold for implementation. Here, as well, our study thus also confirms high-level governance, policy, and infrastructure development as the other key intervention for the urgent and effective advancement of planetary health and sustainable healthcare [51,54,55,56].

Our study was not free from limitations. The professions were not equally represented in both phases of our study. Most of the respondents to the questionnaire were occupational therapists and worked in two of the seven provinces of South Africa, specifically, KwaZulu Natal and Gauteng. As the FGDs were conducted virtually, occasional poor network connectivity led to sections in the recordings that were not sufficiently audible for transcription. Self-reported knowledge is limiting, especially as explanations of some key concepts in the questionnaire items were not given. Additionally, completion of the questionnaire may have seemed too arduous, considering only 100 of the 203 participants (51%) did not complete beyond the demographics section. The first question required a matching of terms and definitions, and in its formulation, this question may have been confusing or rendered participants unsure of how to proceed. It is also possible that our study only included participants that had a pre-existing interest in environmental sustainability and so a disposition to have some basic knowledge and positive attitudes to the subject matter.

We recommend that future research build on the findings of this study while considering its limitations. For a better completion rate of the questionnaire, further instrument validation needs to be performed prior to its roll out in other contexts. Considering the vulnerability of Sub-Saharan Africa we recommend continued research especially in this region, where corresponding knowledge is equally lacking and urgent. Most importantly, we need concerted efforts in education; research that generates reliable and evidence-based suggestions for practice; and the policies and infrastructures that will make environmental sustainability a staple foundation of Sub-Saharan Africa’s healthcare systems and services.

## 5. Conclusions

To lay the foundation for context-relevant interventions to advance environmental sustainability in South Africa’s healthcare system and services, the aim of this study was to identify South Africa’s healthcare professionals’ KAP, and barriers regarding environmental sustainability in healthcare. This was achieved through a convergent mixed-methods study design. Results indicated overwhelmingly positive attitudes and a high degree of interest in education, implementation, and taking on more corresponding responsibility. There is, however, a lack of substantial knowledge on the subject matter, and only tentative implementation of practices relevant to recycling, reusing, and reducing waste and resource consumption. This highlights the need for urgent integration of sustainable healthcare and planetary health education in South Africa’s healthcare professionals’ curricula. Additionally, systemic barriers to implementation—lacking resources (financial, time, and material), policies and infrastructures—must be addressed to harness the extant enthusiasm and advance environmental sustainability in healthcare practice on the African continent.

## Figures and Tables

**Table 1 ijerph-19-10121-t001:** Demographic information.

	Quantitative	Qualitative
**Participants’ Profession**	**Overall (*n* = 100)**	**Overall (*n* = 18)**
Audiology	4	1
Both Speech-Language Pathology and Audiology ^1^	2	0
Dietetics/Human Nutrition ^2^	3	0
Occupational Therapy	69	4
Physiotherapy	18	11
Speech-Language Pathology	4	2
**Participants’ Age**	**Overall (*n* = 100)**	**Overall (*n* = 18)**
Mean (SD)	33.6 (±9.7)	39.9 (±13.3)
Range	22–65	24–71

^1^ Speech-Language Pathology and Audiology were previously one degree at some tertiary institutions. ^2^ Dietetics and Human Nutrition are currently one degree at some tertiary institutions.

**Table 2 ijerph-19-10121-t002:** Primary area of work.

	Quantitative (*n* = 100)		Qualitative (*n* = 18)	
	Count	Percentage (%)	Count	Percentage (%)
Academia	10	6.4	6	33
Acute/secondary care or rehabilitation	14	8.9	6	33
Community care	14	8.9	7	38
Education	24	15.3	4	22
Mental health services	14	8.9	0	0
Non-governmental organiations	4	2.6	2	11
Older persons residential facility	1	0.6	2	11
Private practice	49	31.2	6	33
Tertiary healthcare	9	5.7	5	28
Other	7	4.5	7	39

**Table 3 ijerph-19-10121-t003:** Knowledge (*n* = 100).

Question	Count (Percentage)			
	Yes		No	
In practicing as a healthcare professional, I consider environmental sustainability when using/purchasing equipment, resources, consumables and/or devices	49 (49%)		51 (51%)	
	**No knowledge**	**Limited**	**Good**	**Extensive**
Describe your knowledge on: environmental degradation	6 (6%)	60 (60%)	34 (34%)	0 (0%)
environmental sustainability	1 (1%)	49 (49%)	48 (48%)	2 (2%)
sustainable healthcare	5 (5%)	59 (59%)	34 (34%)	2 (2%)

## Data Availability

The data presented in this study are available on reasonable request.

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
