# Peer review of "South African Healthcare Professionals’ Knowledge, Attitudes, and Practices Regarding Environmental Sustainability in Healthcare: A Mixed-Methods Study"

_ijerph, 2022, doi:10.3390/ijerph191610121_

Round 1

Reviewer 1 Report

Lister et al. have conducted a research study that aims to assess South African healthcare professionals' knowledge, attitudes, and practices regarding environmental sustainability in healthcare by developing questionnaires and conducting conversation groups and interviews. The article is fantastically written, with very detailed sections and very extensive Results and Discussion sections. I only have to highlight some minor aspects to correct before the publication of the work:

- There are some abbreviations that must be described in the text before appearing (KAPs, OT and PT).

- Line 111: "Human nutrition" is repeated.

- Line 139: I think the word "administration" is not correct in this context. I would replace it with "distribution".

- Section 2.2.3.: I think it would be very appropriate to include a copy of the questionnaire used as an Annex or Supplementary Material to the article.

- What has been the specific reason for selecting the type of medical specialties included in the work? I think it should be justified in the paper.

Author Response

Dear reviewer, thank you for your gracious feedback and comments on our article. Please find individual responses for how we dealt with each of the points you have raised listed below. Other amendments were made corresponding to the other reviewer’s comments. Kind regards, the authors.

Review point 1: There are some abbreviations that must be described in the text before appearing (KAPs, OT and PT).

Thank you, this has been amended. Please see line 86 – first time that KAP is described fully, with the abbreviation in brackets. From line 88 this has been changed to ensure only KAP was used when referring to the knowledge, attitudes and practices.

Review point 2: Line 111. "Human nutrition" is repeated.

Thank you, this has been removed.

Review point 3: Line 139: I think the word "administration" is not correct in this context. I would replace it with "distribution".

The word was changed to distributed (please see line 150).

Review point 4: Section 2.2.3.: I think it would be very appropriate to include a copy of the questionnaire used as an Annex or Supplementary Material to the article.

Thank you for this suggestion; the questionnaire that was used for the quantitative data collection has been added as supplementary material.

Review point 5: What has been the specific reason for selecting the type of medical specialities included in the work? I think it should be justified in the paper.

Thank you for the comment, this has now been justified. Under 2.2.2 Participant, the following information was added: “These professions were specifically chosen since they represent the most common members of the allied health care professional team in the South African context (excluding the medical and nursing practitioners).”

Reviewer 2 Report

Dear Authors,

Your article is very interesting, and I am grateful for the opportunity to read it. I find the subject very interesting, and the results of your analysis give a lot of new information.

Reading the text, I found few elements that I think would improve your article.

  1. Abstract has inappropriate structure. I suggest answering the following aspects: - general context - novelty of the work - methodology used (describe briefly the main methods or treatments applied) - main results and related interpretations. 2. Introduction: This section should briefly place the study in a wide context and emphasize why it is relevant carrying out the analysis. It should define the purpose of the work and its significance. In this perspective, this section is too succinct and fails to effectively point out the relevance of your contribution towards the existing literature. Moreover, the authors do not provide at the end of the section the description of the paper structure, which is very useful for readers.
  2. There is not a literature review part in this manuscript and I strongly recommend to insert this chapter. It would be convenient to make clear at this point the specific objectives of the work and the way to reach them.
  3. It is not clear what the purpose of this research is and what methods were used to perform the scientific analysis. I think it would be appropriate to write a brief introduction, albeit. Because it is not entirely clear now what this article is about. Also why so many short paragraphs in methods chapter?
  4. I would recommend forming and highlighting the discussion part. In my opinion, the authors lack deeper discussions and insights in analyzing the problems.
  5. The Conclusions chapter is the weak part of your article. The conclusion section should be a brief summary of article’s aim, methods and findings. However, it is not here.

Good luck

Author Response

Dear reviewer, thank you for your gracious feedback and comments on our article. Please find individual responses for how we dealt with each of the points you have raised listed below. Other amendments were made corresponding to the other reviewer’s comments. Kind regards, the authors.

Review point 1: Abstract has inappropriate structure. I suggest answering the following aspects: - general context - novelty of the work - methodology used (describe briefly the main methods or treatments applied) - main results and related interpretations. 2. Introduction: This section should briefly place the study in a wide context and emphasize why it is relevant carrying out the analysis. It should define the purpose of the work and its significance. In this perspective, this section is too succinct and fails to effectively point out the relevance of your contribution towards the existing literature. Moreover, the authors do not provide at the end of the section the description of the paper structure, which is very useful for readers.

Thank you for your feedback. The abstract has been reworked, however, remained in alignment with the journal guidelines (including the 200-word limit).

Review point 2: There is not a literature review part in this manuscript and I strongly recommend to insert this chapter. It would be convenient to make clear at this point the specific objectives of the work and the way to reach them.

Thank you for your suggestion. Unfortunately, there currently is very limited available literature regarding the KAP of HCP, and none in South Africa (or Southern Africa) that we could find. Hence, a literature review would have to extend itself beyond the scope of the study and would be redundant. The lack of literature in this area is precisely the reason for having to conduct this study, such that a baseline of information can be found from which further studies can be conducted.

Review point 3: It is not clear what the purpose of this research is and what methods were used to perform the scientific analysis. I think it would be appropriate to write a brief introduction, albeit. Because it is not entirely clear now what this article is about.

Thank you for the recommendation. An introduction was added in section 2.1. to clarify the design of the study, as well as the purpose of each phase (qualitative and quantitative).

Also why so many short paragraphs in methods chapter?

We have removed some of the subheadings in the Methods section and combined short paragraphs to improve the style. Thank you for the suggested change. Based on your overall request of an improvement of the research design and the description of the methods, we have rewritten this section ensure overall better clarity. Where inconsistencies were noted between survey and questionnaire, focus group discussion, and focus group interviews, this has been changed to ensure consistency.

Review point 4: I would recommend forming and highlighting the discussion part. In my opinion, the authors lack deeper discussions and insights in analyzing the problems.

Thank you for this feedback. The discussion section has been interrogated by the authors. Since all authors, except for one, are from South Africa, they have sought to ensure the depth of discussions are relevant to the South African context. Three additional aspects have been incorporated, referencing additional literature, to ensure further depth. Please see lines 452, 460 to 461, and 481 to 484.

Review point 5: The Conclusions chapter is the weak part of your article. The conclusion section should be a brief summary of article’s aim, methods and findings. However, it is not here.

Thank you for your comment. The conclusion section has been rewritten to specify the aim, methodology, findings, and recommendations. Please see lines 498 to 511. 

General: Teviewer 2 indicated that the presentation of results can be improved. Therefore, the following changes were made: The first table was not in reference to this sentence – “For the quantitative phase, the majority of the participants are currently working in KwaZulu-Natal (a province in South Africa), and 31% of the participants also obtained their undergraduate degree at the University of KwaZulu-Natal (indicated in Table 1 and Table 2).” The sentences have therefore been removed. We have also interrogated the rest of the presentation and made minor clarification changes (added in brackets and highlighted).

Reviewer 2 also indicated the need for minor spelling changes. These have been incorporated within the text, together with some amendments from the passive to the active voice.

Round 2

Reviewer 2 Report

Accept in present form